# Increased Diagnostic Yield of Array Comparative Genomic Hybridization for Autism Spectrum Disorder in One Institution in Taiwan

**DOI:** 10.3390/medicina58010015

**Published:** 2021-12-22

**Authors:** Chung-Lin Lee, Chih-Kuang Chuang, Ru-Yi Tu, Huei-Ching Chiu, Yun-Ting Lo, Ya-Hui Chang, Yen-Jiun Chen, Chao-Ling Chou, Peih-Shan Wu, Chih-Ping Chen, Hsiang-Yu Lin, Shuan-Pei Lin

**Affiliations:** 1Department of Pediatrics, MacKay Memorial Hospital, Taipei 10449, Taiwan; clampcage@gmail.com (C.-L.L.); g880a01@mmh.org.tw (H.-C.C.); wish1001026@gmail.com (Y.-H.C.); Yenyengu@gmail.com (Y.-J.C.); fishymit@gmail.com (C.-L.C.); 2Institute of Clinical Medicine, National Yang-Ming Chiao-Tung University, Taipei 11221, Taiwan; 3Department of Rare Disease Center, MacKay Memorial Hospital, Taipei 10449, Taiwan; andy11tw.e347@mmh.org.tw; 4Department of Medicine, MacKay Medical College, New Taipei City 25245, Taiwan; 5MacKay Junior College of Medicine, Nursing and Management, Taipei 10449, Taiwan; cpc_mmh@yahoo.com; 6Department of Medical Research, Division of Genetics and Metabolism, MacKay Memorial Hospital, Taipei 10449, Taiwan; mmhcck@gmail.com (C.-K.C.); likemaruko@hotmail.com (R.-Y.T.); 7College of Medicine, Fu-Jen Catholic University, Taipei 24205, Taiwan; 8Gene Biodesign Co., Ltd., Taipei 10682, Taiwan; genebiodesign@gmail.com; 9Departments of Obstetrics and Gynecology, MacKay Memorial Hospital, Taipei 10449, Taiwan; 10Department of Biotechnology, Asia University, Taichung 41354, Taiwan; 11School of Chinese Medicine, College of Chinese Medicine, China Medical University, Taichung 40402, Taiwan; 12Institute of Clinical and Community Health Nursing, National Yang-Ming Chiao-Tung University, Taipei 11221, Taiwan; 13Department of Obstetrics and Gynecology, School of Medicine, National Yang-Ming Chiao-Tung University, Taipei 11221, Taiwan; 14Department of Medical Research, China Medical University Hospital, China Medical University, Taichung 40402, Taiwan; 15Department of Infant and Child Care, National Taipei University of Nursing and Health Sciences, Taipei 11219, Taiwan

**Keywords:** autism spectrum disorder, Taiwan, array-CGH

## Abstract

*Background and Objectives*: Chromosomal microarray offers superior sensitivity for identification of submicroscopic copy number variants (CNVs) and is recommended for the initial genetic testing of patients with autism spectrum disorder (ASD). This study aims to determine the diagnostic yield of array comparative genomic hybridization (array-CGH) in ASD patients from a cohort of Chinese patients in Taiwan. *Materials and Methods*: Enrolled in this study were 80 ASD children (49 males and 31 females; 2–16 years old) followed up at Taipei MacKay Memorial Hospital between January 2010 and December 2020. The genomic DNA extracted from blood samples was analyzed by array-CGH via the Affymetrix GeneChip Genome-Wide Human single nucleotide polymorphism (SNP) and NimbleGen International Standards for Cytogenomic Arrays (ISCA) Plus Cytogenetic Arrays. The CNVs were classified into five groups: pathogenic (pathologic variant), likely pathogenic (potential pathologic variant), likely benign (potential normal genomic variant), benign (normal genomic variant), and uncertain clinical significance (variance of uncertain significance), according to the American College of Medical Genetics (ACMG) guidelines. *Results*: We identified 47 CNVs, 31 of which in 27 patients were clinically significant. The overall diagnostic yield was 33.8%. The most frequently clinically significant CNV was 15q11.2 deletion, which was present in 4 (5.0%) patients. *Conclusions*: In this study, a satisfactory diagnostic yield of array-CGH was demonstrated in a Taiwanese ASD patient cohort, supporting the clinical usefulness of array-CGH as the first-line testing of ASD in Taiwan.

## 1. Introduction

Autism spectrum disorder (ASD) is a neurodevelopmental disorder wherein patients have difficulty in communication and social interactions, stereotypical behaviors, and restricted interests. ASD has a prevalence of 1 in 161 children and is more frequent in males [1]. Its pathogenesis is multifactorial, but genetic alteration is the most important factor, with a heterogeneous change seen across the whole genome [2].

Array comparative genomic hybridization (array-CGH) and single nucleotide polymorphism (SNP) genotyping array, as chromosomal microarray analysis (CMA), are initially performed as cytogenetic diagnostic tests for ASD [3,4]. Before the development of CMA, karyotyping was the standard method to detect genetic anomalies in ASD patients. However, this could only detect large and microscopically visible chromosomal changes (>5–7 Mb), with a low diagnostic rate (3–5%) [3,5]. Fluorescence in situ hybridization (FISH) is another tool for detecting submicroscopic deletions and duplications. It could increase the diagnostic yield by 2% to 3% [3,6,7]. Nevertheless, karyotyping and FISH are not enough for evaluating the genetic etiology of ASD.

CMA can overcome the technical limitations of karyotyping and FISH as well as provide a higher resolution of the genome. The International Collaboration for Clinical Genomics, also known as the International Standard for Cytogenomic Array (ISCA) Consortium, recommends CMA as the first cytogenetic diagnostic test in non-syndromic ASD patients [3,8]. The American College of Medical Genetics (ACMG) also established the guidelines of CMA using [3,9,10]. Two studies have described the diseases related to abnormal findings in CMA. Ellison et al. reviewed 46,298 patients via CMA and found 151 disorders related to chromosomal/genetic abnormalities [3,11], with 35% of the patients having abnormal CMA findings. Riggs et al. surveyed the ISCA Consortium database and found 28,526 patients with 146 phenotypes [3,12]. Among the copy number variants (CNVs), 46% were found to be either pathogenic or likely pathogenic (1908/4125).

Many studies have described the causative role of CNVs in ASD [3,13], congenital heart diseases [3,14], epilepsy [3,15], and congenital kidney malformation [3,16]. However, the same CNVs might cause multiple diseases, and the development of disease can be attributed to many different factors. This is known as the two-hit hypothesis [3,17,18]. Due to the “two-hit hypothesis”, the clinical diagnosis, genetic counseling, and management become challenging.

In this study, we used array-CGH to evaluate ASD patients in Taiwan. The diagnostic rate was detected by array-CGH. We also analyzed the CNV characteristic and feature of these patients.

## 2. Materials and methods

### 2.1. Patients

This study assessed 80 idiopathic ASD children (49 males and 31 females) with ages ranging from 2 to 16 years. These patients were not related to each other. All of them were followed up at Taipei MacKay Memorial Hospital between January 2010 and December 2020. An autism diagnostic interview-revised (ADI-R) [19] was used to confirm the diagnosis of autism. These patients were diagnosed with idiopathic ASD, which is of unknown origin, and we excluded other potential etiologies such as neurocutaneous syndromes, other specific syndromes, and congenital or acquired infections among other common causes of autism before they had array-CGH. The intellectual level information was confirmed by the Wechsler Preschool and Primary Scale of Intelligence Fourth Edition (WPPSI-IV) and Wechsler intelligence scale for children–Fifth Edition (WISC-V) [20,21].

### 2.2. Array-CGH and Data Interpretation

We extracted genomic DNA from the peripheral blood according to standard protocols (Figure 1). All samples were sent to two different laboratories. The first laboratory, the National Center for Genome Medicine in Taiwan, used the Affymetrix GeneChip Genome-Wide Human SNP array 6.0 (Affymetrix, Santa Clara, CA, USA), while the second laboratory, Gene Biodesign, used the NimbleGen ISCA Plus Cytogenetic Array (Roche NimbleGen, Madison, WI, USA). The Affymetrix GeneChip Genome-Wide Human SNP array 6.0 had 50,000, 950,000, and 2,700,000 probes with resolutions ranging from 100 to 200 kb across the entire genome to detect CNVs. The Affymetrix Genotyping ConsoleTM version 3.0.1. was used to analyze the array data of 28 patients in this study. The NimbleGen ISCA Plus Cytogenetic Array contained 630,000 and 1,400,000 probes with a resolution of about 15–30 kb throughout the whole genome. The related data were represented using Nexus 6.1 (BioDiscovery, Hawthorne, CA, USA) for 12 patients in this study [3]. We handled all samples according to the manufacturers’ instructions. SPSS version 25.0 (SPSS, Inc., Chicago, IL, USA) was used to perform the statistical analysis. Statistical significance was set at *p* < 0.05.

According to ACMG guidelines [9,10], CNVs fall under one of the following five categories: pathogenic (pathologic variant), likely pathogenic (potential pathologic variant), likely benign (potential normal genomic variant), benign (normal genomic variant), and uncertain clinical significance (variance of uncertain significance (VOUS)). Pathogenic CNVs are those which cause recognized microdeletion and microduplication syndromes. These CNVs contain morbid Online Mendelian Inheritance in Man (OMIM) genes and large deletions or duplications (usually >3 Mb in size) involving many OMIM genes. They are also inherited from an affected parent and greater than 1 copy number amplification. However, it does not occur in MECP2 duplication where in some instances the parent is not affected [22]. Benign CNVs include those that are well-documented in the normal population or the public databases, not previously reported but inherited from a healthy parent, without any morbid OMIM genes, and duplications with no known dosage-sensitive genes. VOUS CNVs are those that cannot be classified as pathogenic or benign due to insufficient evidence. Recent literature does not recommend using “VOUS” to represent the “likely pathogenic” or “likely benign” categories [9]. Combining the “likely” categories and VOUS may be confusing for clinicians and patients receiving clinical reports. The cut-off value is <1.2 for loss (deletion) and >2.8 for gain (duplication). We compared the findings of our study with previous reports and evaluated the morbidity of the genes by using the following publicly available databases: Database of Genomic Variants (DGV), Database of Chromosomal Imbalance and Phenotype in Humans Using Ensemble Resources (DECIPHER), OMIM, PubMed, ClinVar, and the UCSC Genome Browser. All genomic coordinates are based on the February 2009 assembly of the Genome Reference Consortium build 37(GRCh37)/UCSC hg19.

## 3. Results

Figure 2 illustrates the diagnostic work-up of patients with ASD. A total of 47 CNVs were found in 39 ASD patients. Thirty-one patients had only one CNV and eight patients had two CNVs. Among the 47 CNVs, 32 were deletions and 15 were duplications. These CNVs were classified into the following five groups according to the clinical interpretation: 42.6% (20/47) were classified as pathogenic, 23.4% (11/47) as likely pathogenic, 27.6% (13/47) as VOUS, 0% (0/48) as likely benign, and 6.4% (3/47) as benign.

The summary of patient characteristics and CNV findings is shown in Table 1. There were 47 CNVs and 80 ASD patients. The detection rate of CNVs was 58.8%. In our study, there were 24 males and 15 females with CNVs. In male and female patients, the CNV detection rates were 62.5% and 53.1%, respectively. There were 31 clinically significant CNVs in 27 patients with a diagnostic yield of 33.8%. VOUS were detected in 13 patients (16.3%). We reviewed the detected CNVs according to the published CNV map of the human genome [23].

Among the 47 CNVs, 31 (65.9%) were clinically significant; 13 were duplications and 18 were deletions. The largest and smallest sizes of these significant CNVs were 17.59 Mb and 0.008 Mb, respectively. There were 22 (70.9%) CNVs smaller than 5 Mb that could not be routinely detected by karyotyping. Among the 22 CNVs, 16 (51.6%) were between 1 and 5 Mb, while 6 (19.3%) were <1 Mb.

Table 2 illustrates all clinically significant CNVs (31 CNVs) in our study. Deletions in chromosome band 15q11.2 were detected in 4 patients and these deletions were found mostly in our patients. The chromosome band 15q11.2 overlapped the Prader–Willi/Angelman region and involved the *UBE3A*, *SNRPN*, and *CHRNA7* genes. Table 3 describes all 13 VOUS; 3 duplications and 10 deletions. Their sizes ranged from 0.012–148.290 Mb.

## 4. Discussion

There were 27 ASD patients (33.8%) in our study with clinically significant CNVs detected by array-CGH. The rate of diagnosis is relatively high compared with other studies [24,25]. Our study showed that array-CGH could be the first-tier testing for idiopathic ASD patients due to a satisfactory diagnostic yield. Furthermore, array-CGH allows us to describe the breakpoints (BPs) of the CNV. It can also strengthen the genotype–phenotype correlation and identify candidate genes [4]. For example, a patient who had a 15q11.2 deletion at the Prader–Willi/Angelman region eventually developed autism and language delays due to a reported microdeletion at 15q11.2 between BP1 to BP2 [26,27]. 15q11.2 deletion was also the most common clinically significant CNV identified in our cohort.

In our study, the largest clinically significant CNV was a deletion within the chromosome band 18q21.33q23, and it was consistent with a previous report [28]. The 18q21.33q23 deletion had a size of 17.59 Mb and included 44 OMIM genes from *PHLPP1* to *PARD6G*. According to a previous study, 18q deletion was associated to different phenotypes due to its remarkable genomic heterogeneity [29]. Therefore, we could not confirm diagnosis of 18q deletion by clinical characteristics; genomic analysis is necessary. Our patient had cognition delay, expressive language delay, gross and fine motor delay, hearing loss, delayed myelination of the brain, umbilical hernia, and ear canal stenosis, symptoms compatible with distal 18q deletion [30]. In previous studies, about 54% of the patients with 18q deletion had congenital cardiac anomalies [30,31,32]; however, our patient had a normal echocardiogram. The constitutional hemizygosity of 18q increases the risk of autism as well; 43% of 18q-deletion patients had autism [33]. Furthermore, if the *TCF4*, *NETO1*, and *FBXO15* genes were in the region of hemizygosity, the risk of autism increased significantly [32]. Our patient had deletion of the *NETO1* and *FBXO15* genes. However, there was no shared region of deletion in the ASD patients with 18q deletion. Therefore, further studies are needed to confirm the genetic determinants of autism in 18q-deletion patients.

One patient in our cohort had a 14q21.2q22.1 deletion involving the NIN gene. Microcephalic primordial dwarfism disorder has been associated with compound heterozygous mutations of *NIN* gene [34]. However, our patient had only developmental delays without dysmorphic features. Ninein, a centrosomal protein involved in microtubule anchoring, is encoded by the *NIN* gene. Ninein plays an important role in microtubule stability due its influence in axonal development and bifurcation [35,36]. Disruptions of neocortex development and axon guidance are crucial factors for the development of ASD [37,38,39,40]. Thus, the NIN gene was associated with ASD possibly because of the function of ninein in axonal development and bifurcation.

The differences in certain aspects between Taiwan and European cohorts were noted by the CNV data from other studies [41,42,43,44]. According to previous reports, the most common detected CNVs in ASD occur in 16p11.2; however, this is seen in less than 1% of ASD patients [42,43,44,45]. In our study, the most frequently detected CNVs were 15q11.2 deletions, seen in 5.0% of ASD patients. To evaluate the differences between Taiwan ASD patients and other ASD cohorts, further studies are needed to assess larger Taiwan ASD cohorts compared with controls. On the other hand, VOUS comprised 13 out of 47 (27.7%) CNVs in our study. It is crucial to interpret VOUS in the context of parental data, but this information was not available during data collection. Due to their possible association with ASD, further investigations for VOUS are needed.

In our study, there were 41 patients without CNVs. However, aside from CNVs, other factors like damaging missense mutations, epigenetic alterations, environmental (in utero and early childhood), developmental factors and as-yet unknown different ways influence autism phenotype [46,47]. Based on research to date, a single condition or event could not play a major role in causing ASD. Even though syndromic or secondary autism caused by such as fragile X syndrome and tuberous sclerosis, none of these etiologies are specific to autism because these etiologies include variable proportion of patients with or without ASD [48]. New technologies in genomics and epigenomics research could uncover the epidemiology of ASD [49]. CMA has a higher resolution than conventional karyotyping. However, CMA may miss polyploidy, balanced translocations, inversion, low-level mosaicism, and marker chromosomes. Meanwhile, we could not exclude all genetic diseases by a benign CMA result. Thus, CMA should not replace the karyotyping.

There are more than 800 genes associated with autism according to case-control studies on population and animal models. In addition to three relevant CNVs and their association with ASD above, we also found other clinically significant CNVs in Table 2 and Table 3. These genes are associated with chromatin remodeling and transcriptional regulation, cell proliferation, and mostly synaptic architecture and functionality. According to the largest exome sequencing study of ASD to date [50], these genes were indicated. Other amygdala-expressed genes associated to the social pathophysiology of ASD were indicated by Herrero et al.’s survey [51].

There were some limitations in this study. Compared to the total number of ASD patients in other studies, the number of patients in our cohort was relatively small. Another limitation was that we did not have the parental samples, which could have helped determine the inheritance for VOUS. In addition, our cohort lacked cases of control CNV data from normal individuals. In other previous studies [28,52,53,54,55,56], there were also no control CNV data from normal individuals. However, according to Kousoulidou et al. study, 6 out of 50 mothers and 8 out of 50 fathers from a total of 100 parents (14%) who had ASD children appeared to carry 16 different rare variants associated with ASD [57]. From an analytical aspect, we also did not check the CNV findings using a second method. However, we reviewed all raw CNV data manually, and this matched the recommended quality parameters.

In our study, two kinds of different CMA testing were used. The methodological factors could influence the results due to different reference samples. We should use the same reference sample within one study [58]. Furthermore, according to Dana Hollenbeck et al. in 2017 [59], there are diagnostic clinical relevance of small (<500 kb) nonrecurrent CNVs during CMA clinical testing. It is necessary for careful clinical interpretation of these CNVs. These small, nonrecurrent CNVs can also facilitate the discovery of new genes involved in the pathogenesis of neurodevelopmental disorders and/or congenital anomalies. Our patient with CNV <500 kb, particularly <50 kb, did not have multiplex ligation-dependent probe amplification (MLPA) or FISH for confirmation. It should be modified in the future.

## 5. Conclusions

In the ASD patient cohort in Taiwan, there was a satisfying diagnostic rate by using array-CGH. Array-CGH could detect CNVs in high resolution. Comparing to the karyotyping, array-CGH could make enormous details to describe the genomic alterations in ASD patients. Therefore, array-CGH is useful for initial testing of ASD patients in Taiwan.

## Figures and Tables

**Figure 1 medicina-58-00015-f001:**
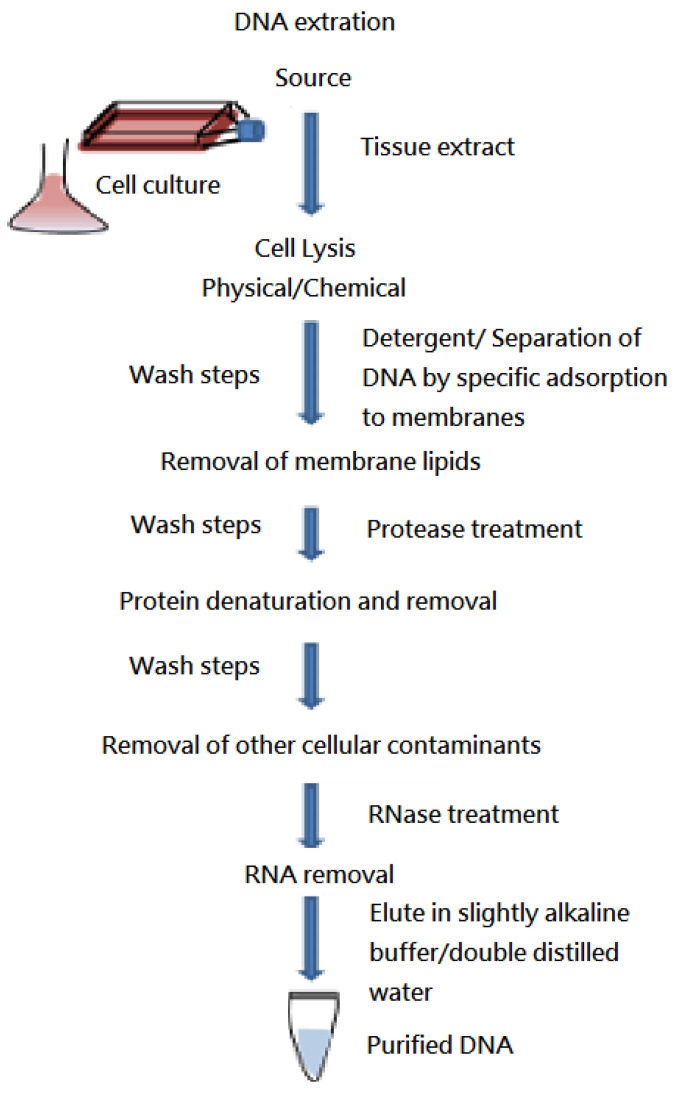
Basic steps involved in all DNA extraction methods.

**Figure 2 medicina-58-00015-f002:**
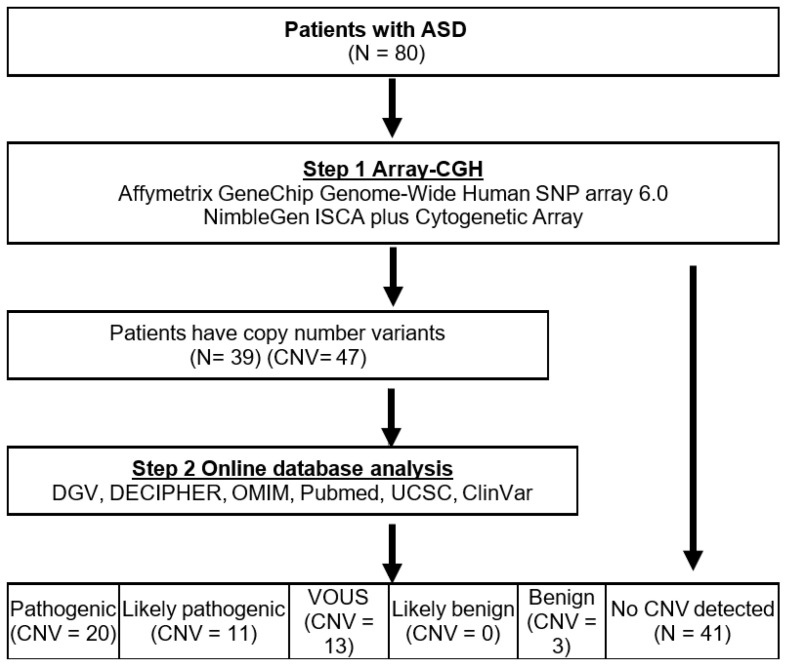
Diagnostic work-up of patients with autism spectrum disorder (ASD) (*N* = 80). Abbreviations: ASD, autism spectrum disorder; CNV, copy number variant; VOUS, variance of uncertain significance.

**Table 1 medicina-58-00015-t001:** Summary of patient characteristics and CNV findings; CNV, copy number variant.

Number of Patients	80
Male	48
Female	32
Age range (median) (years)	2–16 (6)
Total number of CNV	47
(Detection rate %)	(58.8)
Detected in male	30
(Detection rate %)	(62.5)
Detected in female	17
(Detection rate %)	(53.1)
Clinically significant CNV	31
(Diagnostic yield %)	(38.8)
Detected in male	18
(Diagnostic yield %)	(37.5)
Detected in female	13
(Diagnostic yield %)	(40.6)

**Table 2 medicina-58-00015-t002:** Clinically significant CNVs.

Patient Number	Gender	Array CGH Result (hg18)	Chromosome Region (Genes Associated with ASD Phenotype)	Aberration Type	Size (Mb)	Clinical Significance	IQ	Additional Clinical Features
1	Male	arr15q11.2(22,842,145 − 25,235,046) × 3	15q11.2 (*UBE3A*, *SNRPN*, *CHRNA7*)	Duplication	2.393	Susceptibility to ASD	N/A	Developmental delay
		arr15q11.2q13.1(25,236,676 − 28,559,402) × 4	15q11.2q13.1 (*UBE3A*, *SNRPN*, *CHRNA7*)	Duplication	3.323			
3	Male	arr17p11.2(16,782,546 − 20,219,464) × 1	17p11.2 (*RAI1*)	Deletion	3.437	Smith-Magenis syndrome	N/A	Developmental delay and facial dysmorphism
4	Male	arr7q11.23(72,776,313 − 74,133,332) × 1	7q11.23 (*AUTS2*)	Deletion	1.367	Williams syndrome	N/A	Developmental delay
5	Female	arr15q11.2q13.2(22,765,628 − 30,653,876) × 4	15q11.2q13.2 (*UBE3A*, *SNRPN*, *CHRNA7*)	Duplication	7.888	Susceptibility to ASD	N/A	Developmental delay
		arr15q13.2q13.3(30,653,877 − 32,509,926) × 3	15q13.2q13.3 (*CHRNA7*)	Duplication	1.856			
7	Female	arr22q11.21(18,706,001 − 21,505,417) × 3	22q11.21 (*CRKL, FGF8, TBX1*)	Duplication	2.799	Susceptibility to ASD	N/A	Developmental delay and facial dysmorphism
8	Male	arr4p15.1p12(28,451,191 − 47,062,229) × 4	4p15.1p12 (*UGDH*)	Duplication	18.611	Susceptibility to ASD	N/A	Developmental delay
9	Female	arr22q11.23q12.1(25,695,469 − 25,903,543) × 0	22q11.23q12.1 (*CRKL, FGF8, TBX1*)	Deletion	0.208	Susceptibility to ASD	N/A	Developmental delay
11	Male	arr4p16.3(72,447 − 3,848,881) × 1	4p16.3 (*WHS*)	Deletion	3.776	Wolf-Hirschhorn syndrome	33	Developmental delay and facial dysmorphism
12	Female	arr15q11.2q13.3(22,770,421 − 32,915,593) × 1	15q11.2q13.3 (*UBE3A*, *SNRPN*, *CHRNA7*)	Deletion	10.145	Angelman syndrome	N/A	Developmental delay and facial dysmorphism
13	Female	arr4p16.3(68,345 − 4,044,985) × 1.0	4p16.3 (*WHS*)	Deletion	3.977	Wolf-Hirschhorn syndrome	55	Developmental delay and facial dysmorphism
14	Female	arr22q13.33(50,967,018 − 51,197,725) × 1	22q13.3 (*SHANK3*)	Deletion	0.231	Susceptibility to ASD	N/A	Developmental delay and facial dysmorphism
		arr4p16.3p14 (68,345 − 40,111,547) × 3	4p16.3p14 (*WHS*)	Duplication	40.000			
15	Female	arr18p11.32p11.21(136,227 − 15,181,207) × 4	18p11.32 18p11.21 (*SMCHD1*)	Duplication	15.045	Susceptibility to ASD	57	Developmental delay and facial dysmorphism
16	Male	arr11q13.4q14.3(71,567,724 − 89,547,851) × 4	11q13.4q14.3 (*SHANK2*)	Duplication	17.980	Susceptibility to ASD	34	Developmental delay and facial dysmorphism
18	Female	arr2q22.1q22.3(141,332,947 − 145,948,739) × 1	2q22.1q22.3 (*TBR1*)	Deletion	4.161	Susceptibility to ASD	55	Developmental delay and facial dysmorphism
21	Male	arr1p31.3p31.1(61,947,700 − 73,030,143) × 1	1p31.3p31.1 (*NEGR1*)	Deletion	11.080	Susceptibility to ASD	N/A	Developmental delay and facial dysmorphism
22	Female	arr3q22.3q23(138,681,193 − 139,438,715) × 3	3q22.3q23 (*ZBTB20*)	Duplication	0.758	Susceptibility to ASD	N/A	Developmental delay
23	Male	arr10p15.3(162,270 − 468,133) × 3	10p15.3 (*DIP2C*)	Duplication	0.306	Susceptibility to ASD	78	Developmental delay
25	Male	arr14q21.2q22.1(45,863,061 − 50,360,747) × 0	14q21.2q22.1 (*NIN*)	Deletion	4.500	Deletion of the NIN gene	N/A	Developmental delay
27	Female	arr2q23.3q24.1(150,619,633 − 157,576,339) × 1.3	2q23.3q24.1 (*MBD5*)	Deletion	6.957	Susceptibility to ASD	N/A	Developmental delay
28	Male	arr18q21.33q23(60,414,497 − 78,003,508) × 1	18q21.33q23 (*NETO1, FBXO15*)	Deletion	17.590	Susceptibility to ASD	N/A	Developmental delay
29	Male	arr22q11.21(18,657,470 − 21,843,336) × 1	22q11.21 (*CRKL, FGF8, TBX1*)	Deletion	3.190	CATCH22	N/A	Developmental delay
30	Male	arrXp22.31(6,450,627 − 8,141,242) × 0	Xp22.31 (*NLGN4*)	Deletion	1.690	Susceptibility to ASD	80	Developmental delay
		arrXp22.31(8,429,167 − 8,435,863) × 0.5	Xp22.31 (*NLGN4*)	Deletion	1.310			
31	Female	arr15q11.2(20,760,484 − 23,601,857) × 1.1	15q11.2 (*UBE3A*, *SNRPN*, *CHRNA7*)	Deletion	2.840	Susceptibility to ASD	41	Developmental delay
32	Male	arr15q11.2(22,748,697 − 23,188,522) × 1	15q11.2 (*UBE3A*, *SNRPN*, *CHRNA7*)	Deletion	0.440	Susceptibility to ASD	35	Developmental delay
33	Male	arr15q11.2q13.1(23,614,732 − 28,536,497) × 1	15q11.2q13.1 (*UBE3A*, *SNRPN*, *CHRNA7*)	Deletion	4.920	Angelman syndrome	17	Developmental delay
34	Male	arr9q34.3 (140,687,823 − 140,695,906) × 1	9q34.3 (*TSC1*, *EHMT1*)	Deletion	0.008	Kleefstra syndrome	59	Developmental delay and facial dysmorphism
36	Male	arrXq28(152,956,854 − 155,270,560) × 2	Xq28 (*MECP2*)	Duplication	2.310	Susceptibility to ASD	N/A	Developmental delay

N/A, not available; IQ, intelligence quotient; ASD, autism spectrum disorder.

**Table 3 medicina-58-00015-t003:** List of variants of uncertain significance.

Patient Number	Gender	Array CGH Result (hg18)	Chromosome Region (Genes Associated with ASD Phenotype)	Aberration Type	Size (Mb)	IQ	Additional Clinical Features
2	Male	arr22q11.22(22,336,268 − 22,556,733) × 1	22q11.22 (*CRKL, FGF8, TBX1*)	Deletion	0.220	72	Developmental delay
6	Male	arr17p13.3(1693 − 2,393,788) × 1	17p13.3 (*MDLS*)	Deletion	2.392	85	Developmental delay
10	Female	arr9p24.39p23(204,193 − 10,972,824) × 1	9p24.39p23 (*KANK1*)	Deletion	10.768	N/A	Developmental delay
17	Male	arr16q22.1q22.2(69,098,865 − 72,591,930) × 1	16q22.1q22.2 (*SCA4*)	Deletion	3.493	69	Developmental delay
19	Male	arr12p13.33p13.32(173,786 − 4,424,837) × 1	12p13.33p13.32 (*EMG1*)	Deletion	4.250	69	Developmental delay
23	Male	arr20p12.3(8,085,389 − 8,589,571) × 1	20p12.3 (*PLCB1*)	Deletion	0.504	78	Developmental delay
24	Male	arrXq13.1(69,228,881 − 69,240,595) × 0	Xq13.1 (*NLGN3)*	Deletion	0.012	N/A	Developmental delay
26	Female	arrXp21.2(29,336,996 − 29,372,188) × 1	Xp21.2 (*CDKL5*)	Deletion	0.035	N/A	Developmental delay
36	Male	arrXp22.33(1 − 2,196,782) × 0	Xp22.33 (*NLGN4*)	Deletion	2.200	N/A	Developmental delay
37	Female	arr8q21.2q21.13(51,301,121 − 54,915,042) × 1	8q21.2q21.13 (*TCF4*)	Deletion	3.610	19	Developmental delay
38	Male	arrXq13.1q13.3(70,749,306 − 74,335,167) × 2	Xq13.1q13.3 (*NLGN3)*	Duplication	3.590	72	Developmental delay
39	Male	arr17q25.3 (77,856,839 − 78,293,128) × 2.95	17q25.3 (*NF1*)	Duplication	0.436	N/A	Developmental delay
		arrXp22.31q28(6,980,000 − 155,270,000) × 1.1	Xp22.31q28 (*NLGN4*)	Duplication	148.290		

N/A, not available; IQ, intelligence quotient; ASD, autism spectrum disorder.

## Data Availability

All data have been presented in this manuscript.

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
