# Peer review of "Increased Diagnostic Yield of Array Comparative Genomic Hybridization for Autism Spectrum Disorder in One Institution in Taiwan"

_medicina, 2021, doi:10.3390/medicina58010015_

Round 1

Reviewer 1 Report

This is an article to address diagnostic yield of array comparative genomic hybridization for autism spectrum disorder (ASD) in one institution in Taiwan. The authors concluded that a satisfactory diagnostic yield of array-CGH was demonstrated in quite a small (n=80) Taiwanese ASD patient cohort, supporting the clinical usefulness of array-CGH as the first-line testing of ASD in Taiwan. That is interesting and helpful. However, similar content has already been reported, and it lacks originality.

Several concerns were raised as follows:

  1. Manuscript preparation should be constructed according to Instructions for Authors.
  2. Because two kinds of different CMA testing were used, methodological factors that can influence the results should be discussed.
  3. The definition of cut-off value for loss (deletion) and gain (duplication) should be described.
  4. Dana Hollenbeck et al reported the diagnostic clinical relevance of small (<500 kb) nonrecurrent CNVs during CMA clinical testing and underscores the need for careful clinical interpretation of these CNVs. These small, nonrecurrent CNVs can also facilitate the discovery of new genes involved in the pathogenesis of neurodevelopmental disorders and/or congenital anomalies (Genet Med . 2017 Apr;19(4):377-385.). Thus, in this study, patients with CNV < 500 kb should be confirmed by MLPA or FISH, particularly < 50 kb.
  5. Although CMA has a higher resolution than conventional karyotyping, polyploidy, balanced translocations, inversion, low-level mosaicism, and marker chromosomes may be missed. A benign CMA result does not exclude all genetic diseases. Thus, CMA should not replace the karyotyping.

Reviewer 2 Report

Introduction:

Both statements are the same. I should recommend merging by cited together:

"Array comparative genomic hybridization (array-CGH) and single nucleotide poly-
morphism (SNP) genotyping array, as chromosomal microarray analysis (CMA), are ini-
tially performed as cytogenetic diagnostic tests for ASD [3]. Before the development of
CMA, karyotyping was the standard method to detect genetic anomalies in ASD patients.
However, this could only detect large and microscopically visible chromosomal changes
(>5–7 Mb), with a low diagnostic rate (3%–5%) [4]. Fluorescence in situ hybridization
(FISH) is another tool for detecting submicroscopic deletions and duplications. It could
increase the diagnostic yield by 2% to 3% [5,6]. Nevertheless, karyotyping and FISH are
not enough for evaluating the genetic etiology of ASD.
CMA can overcome the technical limitations of karyotyping and FISH as well as pro-
vide a higher resolution of the genome. The International Collaboration for Clinical Ge-
nomics, also known as the International Standard for Cytogenomic Array (ISCA) Consor-
tium, recommends CMA as the first cytogenetic diagnostic test in non-syndromic ASD
patients [7]. The American College of Medical Genetics (ACMG) also established the
guidelines of CMA using [8,9]. Two studies have described the diseases related to abnor-
mal findings in CMA. Ellison et al. reviewed 46,298 patients via CMA and found 151 dis-
orders related to chromosomal/genetic abnormalities [10], with 35% of the patients having
abnormal CMA findings. Riggs et al. surveyed the ISCA Consortium database and found
28,526 patients with 146 phenotypes [11]. Among the copy number variants (CNVs), 46%
were found to be either pathogenic or likely pathogenic (1908/4125). 73
Many studies have described the causative role of CNVs in ASD [12], congenital heart
diseases [13], epilepsy [14], and congenital kidney malformation [15]. However, the same
CNVs might cause multiple diseases, and the development of disease can be attributed to
many different factors. This is known as the two-hit hypothesis [16,17]. Due to “two-hit
hypothesis”, the clinical diagnosis, genetic counseling, and management become challenging."

This text is identical to:

www.pediatr-neonatol.com/articles/S1875-9572(18)30209-2/fulltext/

Round 2

Reviewer 1 Report

The criticism have been generally well revised.